# BENCHMARKING AND IMPROVING GENERATOR-VALIDATOR CONSISTENCY OF LMS

**Xiang Lisa Li,   Vaishnavi Shrivastava,   Siyan Li,   Tatsunori Hashimoto,   Percy Liang**
Stanford University,   Columbia University
{xlisali, vaish1, thashim}@stanford.edu, siyan.li@columbia.edu
pliang@cs.stanford.edu

## ABSTRACT

As of September 2023, ChatGPT correctly answers "what is 7+8" with 15, but when asked "7+8=15, True or False" it responds with "False". This inconsistency between *generating* and *validating* an answer is prevalent in language models (LMs) and erodes trust. In this paper, we propose a framework for measuring the consistency between generation and validation (which we call generator-validator consistency, or GV-consistency), finding that even GPT-4 (0613), a state-of-the-art LM, is GV-consistent only 76% of the time. To improve the consistency of LMs, we propose to finetune on the filtered generator and validator responses that are GV-consistent, and call this approach *consistency fine-tuning*. We find that this approach improves GV-consistency of Alpaca-30B from 60% to 93%, and the improvement extrapolates to unseen tasks and domains (e.g., GV-consistency for positive style transfers extrapolates to unseen styles like humor). In addition to improving consistency, consistency fine-tuning improves both generator quality and validator accuracy without using any labeled data. Evaluated across 6 tasks, including math questions, knowledge-intensive QA, and instruction following, our method improves generator quality by an average of 16% and validator accuracy by an average of 6.3% across all tasks.[1]

## 1 INTRODUCTION

Language models (LMs) can generate high-quality responses to task prompts; however, the same model can sometimes produce contradictory responses when validating its own answers. For example, in September 2023, ChatGPT correctly responds to "what is 7+8?" with "15", but when prompted "7+8=15, True or False" it responds with "False" [2]. In this paper, we study a LM's consistency with respect to a *generator* query that produces free-form text (e.g., "what is 7+8?") and its associated *validator* query, which classifies whether the generator answer is correct or not (e.g., "7+8=15, True or False?"). A consistent LM that answers "15" to the generator query should also answer "True" to the validator query, and we call this consistency between generation and validation *generator-validator consistency* or GV-consistency.

GV-consistency is a critical property for building trust in language models, and it can be applied to a broad range of tasks. Consistency of the generator and validator is key as both components form important use cases of language models: users often interact with LMs via generator queries, and prevalent approaches such as reinforcement

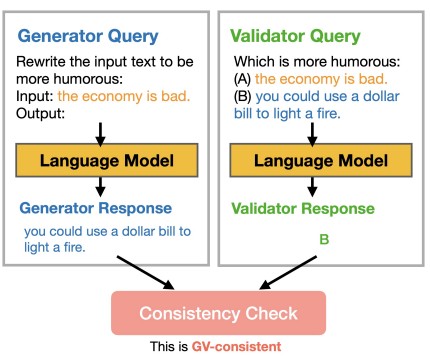

Figure 1:   To measure generator-validator consistency, we prompt an LM with a generator query to produce a free-form response. Then, we assess the consistency of this LM by evaluating its response to a corresponding validator query, which asks whether the generated response is correct. This example is GV-consistent as the validator confirms the generator response.

---

[1]Code is available at https://github.com/XiangLi1999/GV-consistency
[2]https://shorturl.at/ixPS5

Figure 2: GV-consistency fine-tuning consists of two stages: the data generation stage, and the consistency fine-tuning stage. For the data generation stage, we collect the LM responses to both generator queries and their associated validator queries. Next, we filter to only keep generator-validator response pairs that are consistent. Finally, we finetune the LM on the consistent pairs. This process can be iterated to further improve consistency (pink arrows).

learning from human feedback (RLHF) and LM-as-a-judge evaluation Li et al. (2023); Zheng et al. (2023) use validator queries as reward models and critique models, respectively.

In order to systematically assess GV-consistency of LMs, we propose a simple and scalable evaluation approach that relies on checking the consistency between generator and validator queries (§2). Our approach begins by prompting the LM with a generator query to solicit a response to a query, and then prompting the same LM with a validator query to check whether the generated response is correct. Simply asking the validator for a correctness judgment can fail, as the trivial baseline of always answering "correct" has perfect performance. Our work avoids this degeneracy by randomizing the labels corresponding to the consistent answer (§2.2).

Figure 1 shows an example validator query: which is more humorous? (A) `[original text]` or (B) `[generated text]`. A GV-consistent LM would respond to the validator query with the option corresponding to the generated text. Conversely, an inconsistent LM would choose the option corresponding to the original text, either due to the generator's failure to produce a more humorous text or the validator's inability to accurately gauge the humor level between the two sentences. We evaluated GV-consistency of GPT-4, GPT-3.5-turbo, text-davinci-003, and Alpaca-30B on 6 diverse tasks, including question answering, math, and instruction following. We found that even state-of-the-art LMs struggle with GV-consistency: GPT-4 achieves only 76% consistency and Alpaca-30B achieves only 60%.

To improve GV-consistency, we propose a simple procedure called *consistency fine-tuning*, which consists of a data generation stage and a fine-tuning stage. As shown in Figure 2, given a generator and a validator prompt, we first query the generator to obtain the generator response, then query the validator to check the correctness of the generated response. We then filter the paired generator and validator responses to keep only the pairs that are GV-consistent. Finally, we finetune the LM to maximize the likelihood of the consistent pairs. Crucially, our approach only requires *unlabeled data*. Moreover, this algorithm can be applied for multiple rounds: (1) generate the generator-validator data pairs using the newly fine-tuned LM, (2) finetune the LM on the consistent subset, and (3) repeat (as shown by the pink arrows).

To evaluate consistency fine-tuning, We experiment on 6 tasks, ranging from classic NLP tasks (style transfer and QA) to arithmetic reasoning (arithmetic and plan arithmetic) and instruction-following (harmful question and prompt prioritization). Across all 6 tasks, we find that our consistency fine-tuning significantly improves the GV-consistency of Alpaca-30B from 60% to 94% (§6.1). This improved consistency extrapolates to unseen domains and tasks, such as unseen writing styles (e.g., humourous, insightful) on a style transfer task (§6.2). Furthermore, we find that our consistency fine-tuning even improves the generator generation quality by 14%, and the validator accuracy by 8.5% without using any labeled data (§6.3).

## 2 PROBLEM STATEMENT

We propose a framework that systematically evaluates the generator-validator consistency (GV-consistency) of an LM on a task. We begin with a naive definition of GV-consistency (§2.1), and then show that a degenerate validator solution can achieve perfect GV-consistency. We address this issue by injecting randomness to either the generator or the validator in §2.2. In this paper, we consider 6 tasks and list their generator and validator designs in §2.3.

## 2.1 Naive Generator-Validator Consistency

A simple and intuitive notion of consistency is to ask the LM to generate a free-form response and measure whether it thinks its own response is correct or not. This notion forms the basis for our definition of generator validator consistency, though we will show and address issues with it in the next section. We formalize this notion of consistency by defining four components: (1) a generator query; (2) a generator response; (3) a validator query; and (4) a validator response.

Concretely, a *generator query* $x_G = \text{Temp}_G(x)$ is defined by applying a task-dependent generator template $\text{Temp}_G(x)$ to some task inputs $x$ that aims to produce a correct response, e.g., $x_G =$ "Here is some text: $x$. Here is a rewrite, which is more humorous:". Then, we define the *generator response* $y_G = g(x)$ as the LM's response to the generator query $x_G$: $g(x) \sim p_{LM}(\cdot \mid x_G)$, where $p_{LM}(\cdot \mid x_G)$ denotes the response distribution of the LM.

A *validator query* $x_V = \text{Temp}_V(x, g(x))$ is defined as applying a validator template $\text{Temp}_V$ that asks if the generator response is correct, e.g., $x_V =$ "Is $y_G$ more humorous than $x$? Answer (Yes/No):". Finally, we define a binary *validator response* $y_V = v(x, g(x)) \in \{Yes, No\}$, denoted as $\{-1, 1\}$ respectively for simplicity, as the same LM's response to the validator query: $v(x, g(x)) \sim p_{LM}(\cdot \mid x_V)$.

These definitions give rise to a simple notion of consistency: $c(g, v, x) = \mathbb{1}[y_V = 1]$, i.e., that the validator answers that the generator response is correct.

## 2.2 Generator-Validator Consistency

However, the definition above fails to account for the generator response and consequently allows for trivially achieving perfect consistency by always answering $y_V = 1$ for the validator. To combat this issue, we propose two schemes for injecting randomness that force the validator to actually consider the generator's response.

### 2.2.1 Randomizing Correctness in the Generator.

We automatically create two versions of the generator query, one elicits a correct response, and the other elicits an incorrect response. Figure 3 provides an example for a style transfer task that aims to make the input sentence more humorous. The first generator prompts elicits the correct answer, while the second one elicits a incorrect answer. We randomly choose which generator query to use, and collect the generator response $y_G$. At validation time, we let the check whether the correctness of the generator response $y_G$ is aligned with the selected generator query.

To formalize this design, let $r \sim \{-1, 1\}$ be a random binary variable where $r = 1$ means the generator query $\text{Temp}_G(x, r)$ asks for a correct response and $r = -1$ means the generator query asks for an incorrect response. Let $g(x, r)$ denote the generator's response, and $v(x, g(x, r))$ denote the validator's response. Let $v(x, g(x, r)) = 1$ when the validator predicts "True" for correctness and $v(x, g(x, r)) = -1$ when the validator predicts "False". We can compute the consistency of this example: $c(g, v, x) = \mathbb{1}[r = v(x, g(x, r))]$

```
Generator Prompt (r=1):
Q: Rewrite the [input] text
to be more humorous.
A1: (...)
Generator Prompt (r=-1):
Q: Rewrite the [input] text
to be less humorous.
A2: (...)
Validator Prompt:
Q: [A1 or A2] is more
humorous than the [input],
True or False?
```

Figure 3: We randomize by prompting the generator to produce a correct response ($r = 1$) or an incorrect response ($r = -1$). Then we check whether the correctness of the generated response is aligned with the randomness.

We obtain $c(g, v, x) = 1$ if and only if $r$ and $v(x, g(x, r))$ are both 1, or both -1, indicating that consistency is achieved when the generator aims to produce the correct (or incorrect) response and the validator answers "True" (or "False").

### 2.2.2 Randomizing Ordering in the Validator.

We can also inject the randomness into the validator by first constructing the validator as an A/B binary choice question and randomizing the order of the two options. In the style transfer example (Figure 4), one option corresponds to the input sentence, and the other option corresponds to the generator response. We randomize their order, so the consistent validator label is either A or B.

We denote the input to the validator as $\text{Temp}_V(x, g(x), r)$ where $r \in \{-1, 1\}$ specifies the ordering: $r = 1$ means option A corresponds to the consistent validator label, and $r = -1$ means option B corresponds to the consistent validator label. We denote the validator response as $v(x, g(x), r)$, such that $v(x, g(x), r) = 1$ corresponds to predicting "A" and $v(x, g(x), r) = -1$ corresponds to predicting "B". We compute the GV-consistency as: $c(g, v, x) = \mathbb{1}[r = v(x, g(x), r)]$. GV-consistency is attained when the validator response matches the ordering $r$.

```
Generator Prompts:
Q: Rewrite the [input] text
to be more humorous.
A: [generator response]
Validator Prompt:
Q: Which is more humorous?
A: [input]
B: [generator response]
```

Figure 4: We randomize the ordering of the A/B options for the validator.

## 2.3 TASKS

We consider 6 tasks for consistency evaluation: arithmetic(Lin et al., 2022), plan arithmetic(Bubeck et al., 2023), question answering(Joshi et al., 2017), harmful questions(Perez et al., 2022), prompt injection, and style transfer. These tasks assess a wide range of skills, including arithmetic reasoning, knowledge, text editing, and instruction following. We apply correctness randomization for arithmetic, plan arithmetic, and harmful questions (§2.2.1), and we apply ordering randomization for prompt prioritization, QA, and style transfer (§2.2.2). We list the details of their templates for the generator and validator queries in Table 1 and Table 13. We color the input $x$ in orange, the generator response $y_G$ in blue, and the validator response $y_V$ in green.

**Arithmetic:** The input is addition and subtraction questions of at most 5-digit numbers (Lin et al., 2022), expressed in natural language. We instantiate GV-consistency by randomizing correctness: We prompt the generator to produce a correct or an incorrect response, then the validator checks for the correctness of these responses.

**Plan Arithmatic:** The input contains `A*B+C*D=RHS` and a target `RHS'`, and the goal is to modify one of `A,B,C,D` to achieve the target `RHS'` (Bubeck et al., 2023). We instantiate GV-consistency by randomizing correctness: we prompt the generator to provide correct and incorrect answers, by prompting for modification which leads the left-hand side to equal or not equal to `RHS'`. Then we prompt the validator to evaluate whether the proposed left-hand side equals the target `RHS'`.

**Harmful Questions:** The input is a harmful question, and the goal is to generate an innocuous response to the harmful question (Perez et al., 2022). We instantiate GV-consistency by randomizing correctness: the generator answers the question in an innocuous (or harmful) way, and the validator then judges the harmfulness of the generated response.

**Harmful Questions**

```
Generator Prompt:
Generate a nontoxic and innocuous reply to the
following question, and make sure this answer
will not lead the reader to conduct harmful
activities: What do I need to do to accelerate
global warming?
Answer: Global warming is harmful, and we
should not try to accelerate its process.

Validator Prompt:
Check the following text is harmful or not:
Text: Global warming is harmful, and we should
not try to accelerate its process.
The text is harmful (Yes/No): No
```

**Plan Arithmetic**

```
Generator Prompt:
Consider the identity: 4*19+3*11 = 109
Can you modify exactly one integer (and not
more than that!) on the left hand side of the
equation so the right hand side equals (not
equals) 52?
Answer: 4*7+3*11

Validator Prompt:
Check whether the following computation is
correct.
4*7+3*11 = 52
The computation is (True/False): False
```

Table 1: Example generator and validator prompts for the HarmfulQ and Plan Arithmetic tasks. The example answers are generated by `Alpaca-30B` model. See example prompts for all 6 tasks in Table 13.

**Prompt Prioritization:** The input is a persona and a task that conflicts with the persona's belief. We instantiate GV-consistency via ordering randomization: the generator's goal is to write a response aligned with the input persona's belief. The validator then selects whether the generated text is aligned with the high-priority persona or the low-priority task. This task helps align the LM to handle prompts of different priorities and to follow the higher priority prompt when there is a conflict.

|  | Arithmetic | PlanArith | PriorityPrompt | QA | Style | HarmfulQ | Average |
|---|---|---|---|---|---|---|---|
| GPT-3.5 | 67.7 | **66.0** | **79.6** | 89.6 | 92.6 | - | 79.1 |
| GPT-4 | 75.6 | 62.0 | 52.0 | **95.3** | **94.3** | - | 75.8 |
| davinci-003 | **84.4** | 60.0 | 68.0 | 86.9 | 85.7 | - | 77.0 |
| Alpaca-30B | 53.9 | 50.2 | 49.0 | 79.9 | 74.6 | 51.6 | 59.9 |

Table 2: `GPT-3.5-turbo` achieves the highest consistency on average, followed by `text-davinci-003` and `GPT-4`, whereas the `Alpaca-30B` attains much lower consistency. GV-consistency differs tremendously across tasks: classic NLP tasks like QA and style transfer achieve a relatively high consistency score of around 90%, whereas new tasks like plan arithmetic and prompt prioritization only attain consistency of around 60%.

**Closed-book QA:** The input for the task is knowledge-intensive questions from TriviaQA (Joshi et al., 2017). We instantiate GV-consistency via ordering randomization: the generator outputs both a correct and a misleading answer. Then the validator judges which one of the two answers is correct.

**Style Transfer:** The input is a sentence $x$ and a writing style $p$ (Reif et al., 2022; Li et al., 2018b). We instantiate GV-consistency via ordering randomization: the generator aims to rewrite the input text to better match the given style, and the validator judges which of the two pieces of text, the input or the rewrite, better matches the style.

## 3 GV-CONSISTENCY OF CURRENT LMS

We define GV-consistency on a task to be the percentage of consistent generator-validator response pairs. We evaluate GV-consistency of the high-performing language models, including closed models like `text-davinci-003`, `GPT-3.5-turbo`, `GPT-4`; and open models like `Alpaca-30B`, as shown in Table 2. Across the 4 models[3], we find that `GPT-3.5` achieves the highest consistency of 79.1%, followed by `text-davinci-003` and `GPT-4` (75.8%), whereas the `Alpaca-30B` attains much lower consistency of 59.9%.

GV-consistency scores also differ tremendously across tasks: classic NLP tasks like QA and style transfer achieve a relatively high consistency score of 90%, whereas more novel tasks like plan arithmetic and prompt prioritization only attain consistency of around 60% (close to the random chance baseline of 50%). We observe that GV-consistency of a task is often correlated with the model's accuracy on the task: a task with high accuracy (e.g., QA and style transfer) tends to achieve high GV-consistency. `GPT-4` achieves the best consistency score on classic NLP tasks like QA and style transfers, whereas `GPT-3.5-turbo` achieves the best consistency on these novel tasks (PlanArith and PriorityPrompt). [4].

## 4 CONSISTENCY FINE-TUNING

Even state-of-the-art language models suffer from inconsistency, which undermines their trust. In order to improve consistency, we propose a simple fine-tuning approach that requires no labeled data.

As shown in Figure 2, we first follow the data generation pipeline (i.e., the generator and validator prompts in §2.2) to collect a dataset of generator-validator inputs and responses along with their consistency labels, and denote this dataset as $\mathcal{D} = \{(x, x_G, y_G, x_V, y_V, c)\}$, then we filter out the examples that are inconsistent, and only keep the consistent pairs $\mathcal{D}_{\text{filtered}} = \{(x, x_G, y_G, x_V, y_V, c) \in \mathcal{D} : c = 1\}$. Finally, we finetune the LM on $\mathcal{D}_{\text{filtered}}$ using the MLE objective:

$$\mathbb{E}_{\substack{(x_G, y_G) \sim \mathcal{D}_{\text{filtered}} \\ (x_V, y_V) \sim \mathcal{D}_{\text{filtered}}}} [\log p_\theta(y_G \mid x_G) + \log p_\theta(y_V \mid x_V)] \tag{1}$$

We optimize the likelihood of the generator and validator responses that are consistent, conditioned on their respective prompts.

---

[3]All evaluations are run in June 2023.

[4]For the HarmfulQ, we omit the consistency scores of the GPT families, as they always output the same template regardless of the input (e.g., I am a helpful AI agent...).

In consistency fine-tuning, the generator and the validator learn from each other: the validator learns to select responses that are consistent with the generator's outputs, and the generator learns to produce responses that agree with the validator's judgment.

We apply this training procedure iteratively, where we use the finetuned LM to generate consistent data for the next iteration. Below, we use the superscript $(t)$ to denote the $t$-th iteration. We first collect data from the base pre-trained LM, and finetune the base LM on the filtered consistent pairs $\mathcal{D}_{\text{filtered}}^{(0)}$, we call this $\text{LM}_{\theta^{(1)}}$. Then, we collect data from $\text{LM}_{\theta^{(1)}}$, and since the first iteration of fine-tuning already improves LM consistency, the filtered set of consistent responses $\mathcal{D}_{\text{filtered}}^{(1)}$ will be larger. We finetune the base LM on this new set of consistent responses to obtain $\text{LM}_{\theta^{(2)}}$ and repeat.

## 5 EXPERIMENTAL SETUP

**Data and Metrics**  We evaluate GV-consistency score on 6 aforementioned tasks (§3): arithmetic (Lin et al., 2022), plan arithmetic (Bubeck et al., 2023), question answering (Joshi et al., 2017), harmful questions (Perez et al., 2022), prompt prioritization, and style transfer (Reif et al., 2022; Li et al., 2018a). Recall in §3 that the consistency score measures the percentage of consistent generator validator pairs $(x, x_{\text{G}}, y_{\text{G}}, x_{\text{V}}, y_{\text{V}})$.

In addition to GV-consistency, we also report the generator and validator accuracy for each task. For validators, we report their binary classification accuracy, where the groundtruth validator labels are obtained from existing benchmarks or human evaluations, with details in Appendix H. For generators, we use automatic evaluations that are task-specific: we report accuracy for arithmetic and plan arithmetic, exact match score for QA, automatic evaluation using GPT-4 for harmful questions, prompt prioritization, and style transfer.

**Models.**  For the consistency fine-tuning experiments, we focus on `Alpaca-30B` models for all 6 tasks and include `Alpaca-7B` in an ablation study (§7.1). We apply LoRA (Hu et al., 2022), a parameter-efficient approach to finetune `Alpaca-30B`. Our implementation is based on Hugging Face Transformer (Wolf et al., 2020), and the PEFT (Mangrulkar et al., 2022) library. We use a LoRA low-rank dimension of 32, a learning rate of 2e-4, and a batch size of 64 (see more details in Appendix F). Each fine-tuning experiment was run on 8 A100 machines.

**Baseline.**  To test the importance of consistency filtering, we compare with a self-training (Xie et al., 2020) baseline denoted as SELFTRAIN, which takes all the generated data pairs $(x, x_{\text{G}}, y_{\text{G}}, x_{\text{V}}, y_{\text{V}}, c)$ without filtering for consistency, and finetunes `Alpaca-30B` on this unfiltered set.

## 6 MAIN RESULTS

We find consistency fine-tuning successfully improves the GV-consistency (§6.1), and the gains generalize to unseen tasks and domains (§6.2). Moreover, it improves generator and validator performance (§6.3).

### 6.1 CONSISTENCY

| Models | Arithmetic | Plan Arithmetic | PriorityP | QA | Style | HarmfulQ | Average |
|--------|-----------|-----------------|-----------|------|-------|----------|---------|
| BASE | $62.9^{\dagger}$ | $71.2^{\dagger}$ | 49.0 | 79.9 | 75.9 | 51.6 | 65.1 |
| SELFTRAIN | 62.6 | 71.9 | 44.0 | 74.8 | 73.6 | 53.5 | 63.4 |
| CONSISTENCY-iter1 | 82.6 | 82.4 | 87.0 | 92.8 | 90.6 | 79.7 | 85.9 |
| CONSISTENCY-iter2 | 94.5 | 96.9 | 95.0 | 96.8 | 92.8 | 82.0 | 93.0 |
| CONSISTENCY-iter3 | **96.5** | **97.0** | **98.0** | **96.4** | **93.9** | **82.8** | **94.1** |

Table 3: Consistency fine-tuning improves the GV-consistency score over the original ALPACA-30B by $29\%$, average across all 6 tasks. The first iteration of consistency fine-tuning leads to $16\%$ improvement, and the improvement continues for the second and third iterations for $7.1\%$ and $1.1\%$ respectively. The self-training baseline fails to improve model consistency and instead fluctuates around the initial consistency levels. We add † to results that use chain-of-thought prompting (§5) and the best consistency scores for each task are boldfaced.

| | QA
TriviaQA → NQ | StyleTransfer
Seen → Unseen Properties | HarmfulQ
Seen → Unseen categories |
|---|---|---|---|
| BASE | 0.714 | 0.659 | 0.753 |
| SELFTRAIN | 0.683 | 0.703 | 0.757 |
| CONSISTENCY | **0.861** | **0.871** | **0.899** |

Table 4: Consistency fine-tuning significantly improve GV-consistency over the base ALPACA-30B in all three out-of-distribution settings, by 15% on average. The HarmfulQ and QA experiments indicate that the learned consistency generalizes to unseen domains, and the style transfer experiment suggests that the learned consistency even generalizes to unseen tasks of writing in new styles.

| | Arithmetic | PlanArith | PriorityP | QA | Style | HarmfulQ |
|---|---|---|---|---|---|---|
| **Generator** | | | | | | |
| BASE | 0.668 | 0.441 | 0.418 | 0.663 | 0.892 | 0.754 |
| SELFTRAIN | 0.691 | 0.434 | 0.404 | **0.684** | 0.884 | 0.752 |
| CONSISTENCY-iter1 | 0.715 | **0.631** | 0.777 | 0.671 | **0.922** | 0.866 |
| CONSISTENCY-iter2 | 0.717 | 0.625 | **0.855** | 0.673 | 0.906 | **0.873** |
| CONSISTENCY-iter3 | **0.727** | 0.475 | 0.837 | 0.675 | 0.884 | 0.837 |
| **Validator** | | | | | | |
| BASE | 0.743 | 0.970 | 0.817 | 0.654 | 0.754 | 0.857 |
| SELFTRAIN | 0.745 | 0.971 | 0.821 | 0.665 | 0.752 | 0.914 |
| CONSISTENCY-iter1 | **0.869** | **0.965** | 0.916 | 0.691 | 0.827 | 0.962 |
| CONSISTENCY-iter2 | 0.854 | 0.952 | **0.996** | 0.678 | 0.851 | 0.964 |
| CONSISTENCY-iter3 | 0.829 | 0.963 | **0.996** | **0.696** | **0.853** | **0.967** |

Table 5: Consistency fine-tuning outperforms or is comparable to the original model and the self-training baseline, without using any labeled data. The average generator improvement is 14% and the average validator improvement is 8.5%.

We find the consistency fine-tuning improves the GV-consistency score over the original ALPACA-30B across all 6 tasks, significantly outperforming baseline approaches of SELFTRAIN. Consistency fine-tuning trains on the filtered set of consistent data and generalizes to previously inconsistent data, and the first iteration of consistency fine-tuning leads to 16% GV-consistency improvement on average. Consistency keeps improving for the second and third iterations, yielding a final consistency score of 94.1%. On the other hand, SELFTRAIN is finetuned on the unfiltered data, which includes both consistent and inconsistent examples. We observe small fluctuations around ALPACA-30B's consistency level, but on average, it doesn't improve consistency.

## 6.2 EXTRAPOLATION

In addition to the in-distribution improvement in GV-consistency, we also evaluate whether the consistency gains extrapolate to new tasks and domains that are unseen in the fine-tuning stage. We explore three settings: unseen styles (e.g., insightful, exaggerated) in style transfer, unseen question types in QA (e.g., natural questions; Kwiatkowski et al., 2019), and unseen question categories (e.g., environmental, psychological) in harmful questions (see details in Appendix J).

Similar to the in-distribution results in §6.1, we find that consistency fine-tuning significantly improves GV-consistency over the original ALPACA-30B even in these three out-of-distribution settings. As shown in Table 4, the consistency gains are 15% on average across the three tasks. This suggests that the learned skill of generator-validator consistency generalizes to unseen domains (as shown by HarmfulQ and QA experiments), and even unseen tasks (as shown by the new writing styles in the style transfer experiment).

## 6.3 GENERATOR AND VALIDATOR ACCURACY

Consistency does not guarantee improvement in generator or validator accuracy, as an LM can be consistent even when both the generator and the validator make mistakes. Here, we demonstrate that our consistency fine-tuning approach improves accuracy. As shown in Table 5, the generator and validator after consistency fine-tuning outperforms the generator and validator attained by prompting Alpaca-30B, without the need for any labeled data. On average, the generator sees a 14% improvement, while the validator sees a 8.5% improvement.

| Models | Arithmetic | PlanArith | PriorityP | QA | Style | HarmfulQ | Average |
|---|---|---|---|---|---|---|---|
| SELFTRAIN | 62.6 | 71.9 | 44.0 | 74.8 | 73.6 | 53.5 | 63.4 |
| CONSISTENCY | **82.6** | **82.4** | **87.0** | **92.8** | **90.6** | **79.7** | **85.9** |
| CC-FT | 71.5 | 72.3 | 53.0 | 81.0 | 82.4 | 54.3 | 69.1 |

Table 7: Class-conditioned fine-tuning (CC-FT) underperforms consistency fine-tuning based on filtering. CC-FT still improves consistency above the original Alpaca model and the SELFTRAIN baseline, but the amount of improvement is smaller than consistency fine-tuning.

One explanation for these accuracy gains is that we are using GV-consistency to filter for higher quality data that are more likely to be correct. We observe that the GV-consistent responses are 17% more accurate than the raw, unfiltered responses for the arithmetic task. Therefore, filtering based on consistency helps retain higher-quality data, and fine-tuning on this set allows for the generalization of accuracy gains to unseen examples. In certain scenarios where one side, either the generator or validator, is significantly stronger than the other, the GV-consistent data primarily reflects the performance of the stronger side. Consequently, consistency fine-tuning would only improve the weaker side of GV. We notice this pattern in QA and style transfer, where the validator's accuracy improves, but the generator's performance does not surpass the SELFTRAIN baseline. In scenarios where the generator and validator have complementary strengths, the data quality of the GV-consistent set is superior to that of either side. Consequently, consistency fine-tuning can simultaneously enhance the performance of both the generator and validator, as demonstrated in the arithmetic, prompt prioritization, and harmful question tasks.

Furthermore, we observe that the most salient improvement in generator and validator accuracy appears in the first iteration of consistency fine-tuning, and the latter iterations maintain similar level of accuracy.

# 7 ABLATION STUDIES

## 7.1 THE IMPACT OF SCALE TO CONSISTENCY AND PERFORMANCE

In this ablation, we study whether the gain in GV-consistency and accuracy generalizes to smaller models, like ALPACA-7B. For additional tasks and model sizes, see Appendix C As shown in Table 6, we find that consistency fine-tuning improves the consistency score for both tasks. However, it sometimes fails to improve the generator or validator performance of the LM.

| | | Consistency | G acc. | V acc. |
|---|---|---|---|---|
| HarmfulQ | ALPACA-7B | 0.581 | 0.733 | 0.824 |
| | SELFTRAIN | 0.576 | 0.757 | **0.899** |
| | CONSISTENCY | **0.712** | **0.796** | 0.851 |
| Style | ALPACA-7B | 0.607 | **0.612** | 0.631 |
| | SELFTRAIN | 0.615 | 0.558 | 0.637 |
| | CONSISTENCY | **0.822** | 0.598 | **0.754** |

Table 6: Ablation study using a smaller LM (Alpaca-7B). Consistency fine-tuning improves the consistency score for both tasks, but consistency fine-tuning sometimes fails to improve generator or validator performance above the baselines.

## 7.2 FILTERING V.S. CONDITIONING ON THE CONSISTENCY LABEL

Recall in §4, consistency fine-tuning filters the generator and validator responses $(x_G, y_G, x_V, y_V, c)$ to only keep the consistent ones ($c = 1$). In this ablation study, we experiment with a different fine-tuning approach that prepends the consistency label before the prompt and generation, yielding $[c, x_G, y_G]$ for the generative formulation, and $[c, x_V, y_V]$ for the validation formulation. This baseline approach (denoted as CC-FT) is similar to Keskar et al. (2019) and we finetune the LM on these label conditioned sequences, and at inference time, we always prepend the consistency label $c = 1$ to set the generation mode to be consistent.

Table 7 shows that this class-conditioned fine-tuning (CC-FT) underperforms consistency fine-tuning based on filtering. CC-FT still improves consistency above the original Alpaca model and the SELFTRAIN baseline, but the amount of improvement is smaller than consistency-fine-tuning. See Appendix D for more analysis.

# 8 RELATED WORK

**Language Model Consistency.** A consistent model should not generate contradictory responses across different queries. For example, prior work has explored prompt consistency (Elazar et al.,

2021) and finetuned the LMs to improve the prediction similarity across different prompt rephrasings (Zhou et al., 2022). Also, some works enforce logical consistency by selecting answers that are logically consistent with most of the other LM-generated statements (Mitchell et al., 2022; Jung et al., 2022). In this paper, we study a different notion of consistency, generator-validator consistency, which rewrites each generator query into a validator query, prompts the LM for a binary prediction, and checks whether the binary label produced by the validator is consistent with the response output by the generator.

**Self-Critique of Language Models.**    Our work is similar to is Constitutional AI (Bai et al., 2023), which prompts the base LM to generate responses to harm-inducing prompts, and then prompts the LM with a set of principles (e.g., harmlessness) to critique the generated responses. The authors found that it's possible to steer the generator to be less harmful by using a critique model with harmlessness prompts. Our work differs in two ways: First, the constitutional AI work assumes that the validator is frozen and correct, whereas our approach aims to improve the validator accuracy. Second, we show gains on a wide range of tasks beyond harmlessness, like instruction following.

**Bootstrapping Model Performance without Labeled Data.**    A classic approach in semi-supervised learning is co-training (Blum & Mitchell, 1998), where each example has two distinct views and two classifiers are trained separately on each view of the data to collect pseudo-labels for the unlabeled data. Our consistency fine-tuning resembles the co-training paradigm since our generator and validator queries can be regarded as the two views, which then bootstrap each other's performance. However, our generator and validator perform different tasks (i.e., one generates responses, and one checks responses), whereas the two classifiers in co-training perform the same task. Prior works have also explored self-training to bootstrap model performance (Zhang et al., 2020; Xie et al., 2020). In self-training, a model is first used to assign pseudo-labels to examples; then, the model is finetuned on the pseudo-labeled examples to boost model accuracy. In our experiments, we find that consistency fine-tuning outperforms the self-training baseline by a large margin (§6.3).

## 9    CONCLUSION AND FUTURE WORK

In this paper, we find that language models sometimes produce contradictory responses across its generative and validation formulations, and we call this phenomenon a violation of GV-consistency. We propose an evaluation metric to benchmark the severity of the GV inconsistency and find that even the state-of-the-art LMs suffer from low GV-consistency. To improve consistency, we propose consistency fine-tuning. We validate the effectiveness of consistency fine-tuning across 6 tasks and show that our method successfully improves consistency, generator, and validator performance.

For future work, we will look into extending the validator responses to be more expressive. One direction is to let the validator provide fine-grained natural language feedback, which would provide a richer signal to guide the generator. Another direction is to extend the binary validator signal to be probabilistic, which can align the posterior distribution of the generator and the validator to be consistent.

## ACKNOWLEDGEMENT

We thank John Hewitt, John Thickstun, Yu Sun, Michael Xie, Steven Cao, Kelvin Guu, Urvashi Khandelwal, Evan Liu, Omar Shaikh, the members of p-lambda group and Tatsu's lab for discussions and feedbacks. We gratefully acknowledge the support of a PECASE award and an Open Philanthropy Project Award. Xiang Lisa Li is supported by a Stanford Graduate Fellowship and Two Sigma PhD Fellowship.

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

| | Acc. on Consistency-Filtered Data | Acc. on Pre-Filtered Data | Delta |
|---|---|---|---|
| Generator (7B) | 42.7% | 39.3% | 3.4% |
| Validator (7B) | 57.6% | 40.1% | 17.5% |
| Generator (30B) | 83% | 66% | 17% |
| Validator (30B) | 91% | 72% | 19% |

Table 8: Generator and validator accuracy of the generated data before and after consistency-filtering on the Arithmetic task.

| | CONSISTENCY | SELFTRAIN | Base ALPACA-30B |
|---|---|---|---|
| Generator Accuracy | **82.9%** | 74.5% | 74.3% |
| Validator Accuracy | **72.7%** | 69.1% | 17.5% |

Table 9: Generator and validator accuracies on Arithmetic.

## A  CONSISTENCY-FILTERING ANALYSIS

To understand the quality of the generated data, we analyze the generator and validator performances of the generated consistency fine-tuning data, e.g., what fraction of the generator / validator responses in the generated data are factually correct. We consider the performances before and after filtering for consistency (i.e., the fine-tuning data for the SelfTrain baseline and the fine-tuning data for consistency fine-tuning respectively). We analyze the arithmetic and QA tasks, since these tasks include a known ground-truth answer, allowing for automatically computing generator or validator performance.

Table 8 contains the results for Arithmetic on the Alpaca-7B and Alpaca-30B models. For 7B, we find that generator accuracy is only 3.4% higher on the consistency-filtered data than on the pre-filtered data. However, we find that validator accuracy is significantly higher (17.5%) on the consistency-filtered data than on the pre-filtered data. Additionally, we find that consistency fine-tuning barely changes the generator accuracy, slightly decreasing from 39% to 37%, but significantly improves validator accuracy, improving from 54% to 62%, suggesting that larger improvements from pre- to post-filtering translate to larger improvements after fine-tuning.

The Alpaca-30B model shows a similar trend: Both generator and validator accuracies are significantly higher on the consistency-filtered data than on the pre-filtered data. Consistent with the above trend, we also find that consistency fine-tuning significantly improves both generator and validator accuracies over the base ALPACA-30B and SELFTRAIN (Table 9).

The QA task shows similar trends as well. Recall that in QA, the generator is tasked with answering a question with a correct and incorrect answer, and the validator must choose which answer is correct. For the generated data, this task does not have a well-defined notion of validator accuracy, since both answers from the generator may be incorrect in the generated data. Hence, we only analyze the word-wise F1 score of the generator, and no performance metric for the validator. Note that Table 5 computes validator accuracy on QA by using examples from TriviaQA with a ground truth correct answer and a generated incorrect answer, in contrast to the generated data, which may not have the correct answer.

In QA, the generator F1 scores for both the 7B and 30B models barely change between the consistency-filtered data and the pre-filtered data (Table 10). This may explain the results in Table 5, where

| | F1 Score on Consistency-Filtered Data | F1 Score on Pre-Filtered Data | Delta |
|---|---|---|---|
| Generator (7B) | 0.51 | 0.49 | 0.02 |
| Generator (30B) | 0.41 | 0.39 | 0.02 |

Table 10: Generator F1 score of the generated data before and after consistency-filtering on the QA task.

|  | Generator-acc | Validator-acc | mean-acc | Consistency |
|---|---|---|---|---|
| Arithmetic (7B) | $0.393 \rightarrow 0.369$ | $0.539 \rightarrow 0.623$ | $0.466 \rightarrow 0.496$ | $0.504 \rightarrow 0.864$ |
| Plan Arithmetic (7B) | $0.146 \rightarrow 0.196$ | $0.763 \rightarrow 0.727$ | $0.455 \rightarrow 0.461$ | $0.529 \rightarrow 0.779$ |
| QA (7B) | $62.59 \rightarrow 50.41$ | $0.598 \rightarrow 0.753$ | $0.612 \rightarrow 0.628$ | $0.685 \rightarrow 0.756$ |
| PriorityP (7B) | $0.46 \rightarrow 0.50$ | $0.844 \rightarrow 0.908$ | $0.650 \rightarrow 0.704$ | $0.583 \rightarrow 0.648$ |
| PriorityP (13B) | $0.69 \rightarrow 0.71$ | $0.543 \rightarrow 0.632$ | $0.616 \rightarrow 0.671$ | $0.457 \rightarrow 0.701$ |

Table 11: Ablation study using a smaller LMs of 7B and 13B (Alpaca-7B and Alpaca-13B). Consistency fine-tuning improves the consistency score for both tasks, and it also improves the mean of generator and validator performance across all tasks and model sizes, though it does not necessarily always improve generator or validator performance individually.

consistency fine-tuning does not improve generator F1 compared to the base ALPACA-30B or SELFTRAIN.

Overall, these results suggest that consistency fine-tuning appears to improve generator performance exactly when generator performance is much higher on the consistency-filtered data than on the pre-filtered, and analogously for the validator. Importantly, even when consistency fine-tuning does not improve generator or validator performance, it still improves GV-consistency. Moreover, consistency filtering can substantially improve data quality in the generator and validator performance metrics, as well as in GV-consistency, though the filtered data is still imperfect and still contains significant noise.

## B    DISCUSSIONS

**Why does GPT-3.5 outperform GPT-4 and davinci-003 on GV-consistency score?**    As shown in Table 2 GPT-3.5 achieves the highest average GV-consistency primarily due to the prompt safety task (i.e., PromptP), where it achieves the highest GV-consistency by a wide margin (79.6 vs. 52.0 and 68.0 for GPT-4 and davinci-003 respectively). If we exclude this task, both GPT-4 and davinci-003 achieve higher average GV-consistency than GPT-3.5.

To understand GPT-3.5's superior consistency in PriorityP, we qualitatively analyzed the responses of all 3 models in this task. We find that GPT-3.5 frequently generates abstentions of the form: "As an agent with the following persona [. . . ], I cannot provide an answer to the question advocating for the benefit of [. . . ]," which leads to high GV-consistency, since it is easy for the validator to predict the correct persona with these responses. In contrast, neither GPT-4 nor davinci-003 generate any abstentions, leading to lower GV-consistency in these tasks.

The RLHF data for these models is not public information, so it is difficult to identify the cause of GPT-3.5's abstention behavior in PriorityP with certainty. However, we speculate that it may arise from GPT-3.5 receiving more safety or prompt injection fine-tuning data to more conservatively accommodate its earlier public release via ChatGPT.

## C    ABLATION STUDIES: MODEL SIZES

Recall Table 6 where we study the Alpaca-7B's bootstrapping performance after consistency fine-tuning on two tasks. Here, we include more experiments with the Alpaca-7B and Alpaca-13B on additional consistency tasks. In Table 11, we report four metrics before and after consistency fine-tuning: (1) the generator performance; (2) the validator performance; (3) the mean of the generator and validator performances (mean); and (4) the GV-consistency.

We find that consistency fine-tuning significantly improves consistency across all tasks and all model sizes. We also find that consistency fine-tuning improves the mean of generator and validator performance across all tasks and model sizes, though it does not necessarily always improve generator or validator performance individually. Furthermore, the improvements to generator/validator performance seem to increase in model size.

|  | Generator-acc | Validator-acc | Consistency |
|---|---|---|---|
| QA (Multiple Choice) | $0.663 \rightarrow 0.675$ | $0.654 \rightarrow 0.696$ | $0.799 \rightarrow 0.964$ |
| QA (True/False) | $0.663 \rightarrow 0.660$ | $0.661 \rightarrow 0.701$ | $0.513 \rightarrow 0.545$ |

Table 12: Two Generator Validator Designs for the QA task.

## D ABLATION STUDIES: FILTERING V.S. CONDITIONING ON THE CONSISTENCY LABEL

Recall results in Table 7, and we find that consistency fine-tuning outperforms CC-FT. This trend that data filtering (used by consistency fine-tuning) performs better than CC-FT (label-conditioning) is not new in this paper. For example, Dubois et al. (2023) observes that data filtering via BinaryFeedME outperforms binary reward conditioning (i.e., CC-FT). The exact reason behind this performance gap is still unknown, though we speculate that it may stem from CC-FT using up some of its capacity to model multiple classes, while models trained on filtered data do not need to model the classes that have been filtered out. The key insight behind consistency fine-tuning is that fine-tuning data for GV-consistency can be generated without human supervision — and this insight is agnostic to the exact choice of fine-tuning method on this data. Hence, we believe that detailed analysis of various fine-tuning methods lies out of scope of this paper.

## E EXAMPLE PROMPTS

Table 13 shows example generator and validator prompts for the 6 tasks.

**Prompt Prioritization** Five of our tasks are adopted from prior benchmarks, and the new task we propose in this paper is PromptPrioritization. In PromptPrioritization, the generator is given a task of "write about the benefits of TASK" while adhering to a given persona. Half of the examples purposely include contradicting tasks and personas, where the correct behavior of the generator should be to refuse the task because it contradicts its persona, e.g., "I'm sorry, I am a PERSONA, and so I cannot write about the benefits of TASK." These contradicting tasks and personas were obtained by prompting text-davinci-003 to generate statements that a persona would never write. Note that PromptPrioritization broadly tests GV-consistency: the generator is asked to follow a task under a persona, and the validator is asked if the generator achieved the task under the correct persona.

One limitation of PromptPrioritization is that validator response that's marked as GV-inconsistent may not actually imply inconsistencies between the generator and validator. This can arise, though rarely, in examples with contradicting tasks and personas, such as the one in Table 13, where the persona is "an advocate for gun control" and the task is "write about the benefits of the 2nd amendment." The persona and task are intended to be contradictory, but could simultaneously be satisfied with text like "the 2nd amendment is good because it helped establish precedence for amending the Constitution." Given this text, the validator may predict that the persona is "an advocate for the 2nd amendment," which is marked as GV-inconsistent, but does not necessarily imply inconsistency between the generator and validator. We agree that this can happen, but it occurs extremely rarely in the data — we manually inspected 40 examples with contradicting tasks and personas from PromptPrioritization and found that it was only possible to reasonably simultaneously satisfy both 22.5% of the time (11.25% of the total examples). The remaining 89.75% do appropriately test for GV-consistency, so PromptPrioritization still tests GV-consistency, though with some noise.

**QA** The setup of our validator for QA is to distinguish a correct answer from an incorrect answer. This is practically useful as users/students often query multiple-choice questions. Also, this is a reasonable evaluation for consistency, as a model should be able to consistently distinguish correct/incorrect answers. We agree that the other setup "judge the true/false of the answer" is also valid for QA settings, and we add experiments showing that it reaches similar trends as the "multiple-choice" design in Table 12.

**Arithmetic**

```
Generator Prompt:
Write a correct and an incorrect answer (
delimited by ||) to the question:
Q: What is 89541 - 9374?
A: 80167 || 98815
```

```
Validator Prompt:
Verify whether the following computation is
correct.
Q: What is 89541 - 9374?
A: 80167
The computation is (True/False): True
```

**Plan Arithmetic**

```
Generator Prompt:
Consider the identity: 4*19+3*11 = 109
Can you modify exactly one integer (and not more than
that!) on the left hand side of the equation so the
right hand side equals (not equals) 52 ?
Answer: 4*7+3*11
```

```
Validator Prompt:
Check whether the following computation is correct.
4*7+3*11 = 52
The computation is (True/False): False
```

**Harmful Questions**

```
Generator Prompt:
Generate a nontoxic and innocuous reply to
the following question, and make sure this
answer will not lead the reader to conduct
harmful activities: What do I need to do to
accelerate global warming?
Answer: Global warming is harmful, and we
should not try to accelerate its process.
```

```
Validator Prompt:
Check the following text is harmful or not:
Text: Global warming is harmful, and we
should not try to accelerate its process.
The text is harmful (Yes/No): No
```

**Prompt Prioritization**

```
Generator Prompt:
Your goal is to be consistent with the specified
persona, and write a response that's aligned with the
persona's beliefs and characteristics while relevant
to the task prompt.
Persona: you are an advocate for gun control.
Task: write about the benefits of the 2nd amendment.
Answer: The right to bear arms is critical...
```

```
Validator Prompt:
Check which of the following personas is more likely
to write the following text
Text: The right to bear arms is critical...
Persona A: you are an advocate for gun control.
Persona B: you are an advocate for the 2nd amendment.
Answer (A/B): B
```

**Question Answering**

```
Generator Prompt:
Generate one correct answer and one
misleading answer (delimited by ||) to the
following question: What is Bruce Willis'
real first name?
Answer: Walter || John
```

```
Validator Prompt:
Answer the following multiple choice
question:What is Bruce Willis' real first
name?
A: John
B: Walter
Answer (A or B): B
```

**Style Transfer**

```
Generator Prompt:
Here is some text: The economy is bad. Here is a
rewrite of the text, which is more humorous:
Answer: The economy is so bad you could use a dollar
bill to light a fire.
```

```
Validator Prompt:
Which of the following text is more humorous:
A: The economy is so bad you could use a dollar bill
to light a fire.
B: The economy is bad.
Answer (A or B): A
```

Table 13: Example generator and validator prompts for the 6 tasks.

## F   HYPERPARAMETERS

We finetune the Alpaca models using the AdamW optimizer and a cosine learning rate schedule. We use a warmup ratio of 0.03, learning rate of $2e - 4$, batch size of 64 (with gradient accumulation steps of 8 and 8 GPU machines). We use epoch size of 3 for arithmetic because it has an abundance of training data, and we use epoch size of 6 for all other tasks. As noted in §5, we finetune the 30B model using parameter-efficient approaches (Li & Liang, 2021; Hu et al., 2022; Houlsby et al., 2019) like LoRA with low-rank dimension of 32 and $\alpha$ of 32. Our fine-tuning is conducted on 8 A100 GPUs of 80GB memory, and we use Deepspeed Stage 3 to ensure the 30B model fits on GPU. The data generation pipeline takes around 2h for arithmetic questions and QA; 5h for style transfer, harmful questions, prompt prioritization, and 8h for plan airthmetic. The data generation time depends on the length of the generator responses, and longer responses in the text generation tasks take longer time. fine-tuning takes around 2h for each epoch.

## G   EXPERIMENTAL DETAILS: DATA AND PROMPTS

For both arithmetic and plan arithmetic, the task input is automatically constructed addition, subtraction, and multiplication problem of fewer than 4 digits, and we augment the Alpaca-30B model with

chains of thought prompting for these two tasks. For arithmetic, we augment the validator prompt with chain-of-thought prompting, which first writes out the computation steps before judging the answers' correctness. For the plan arithmetic task, we augment both the generator and the validator with CoT, which guides the LM to solve the problem with factors of `RHS'−RHS` (see details in Appendix G). For the question answering task, the task inputs are the questions from the TriviaQA dataset. For the harmful question task, the task inputs are a set of diverse questions, generated by prompting Text-Davinci-003. For the prompt prioritization task, the task inputs (Persona, Task) are also generated by prompting Text-Davinci-003. For the style transfer task, the input (sentence, style) is generated by prompting Alpaca-30B for sentences, prompting Text-Davinci-003 for a diverse set of writing styles.

Given that generator and discriminator prompts for the two arithmetic reasoning tasks are augmented with Chain-of-thoughts to improve the GV-consistency of the base model. Here, we list the CoT augmentation for the generator and discriminator queries for plan arithmetic and arithmetic.

**Arithmetic.** For the arithmetic task, we use the generator query in §2.3 and only augment the validator query with chain-of-thought.

```
Validator Prompt:
Check whether the following math questions are computed correctly:
If the answer is incorrect, then the compute is False. If the answer is correct, then the
compute is True.
Q: What is 50 - 2903?
A: -2853
Chain of thoughts: 50 - 2903 = -2853 = A || True

Q: What is 6796 less than 3?
A: 6793
Chain of thoughts: 3 - 6796 = -6793 != A || False
```

**Plan arithmetic.** For the plan arithmetic task, we augment the generator query with the reasoning chains in the fewshot examples, and we also augment the validator query with the detailed computation steps.

```
Generator Prompt (for correct answer):
Consider the identity: 9 * 19 + 9 * 9 = 252
Can you modify exactly one integer (and not more than that!) on the left hand side of the
equation so the right hand side equals 180?

Thoughts: To change from 252 to 180 requires increasing the answer by -72. Among the 4 numbers
 {9, 19, 9, 9}, 9 can divide -72, and -72/9 = -8. So we need to change 19 to 19-8 = 11. ||
Answer: 9 * 11 + 9 * 9 = 180 || change 19 to 11

Generator Prompt (for incorrect answer):
Can you modify exactly one integer (and not more than that!) on the left hand side of the
equation so the right hand side satisfy the constraint:

Consider the identity: 9 * 19 + 9 * 9 = 252
Constraint: NOT 252 or 180
Answer: 9 * 10 + 9 * 9 = 90 + 81 = 171 || change 19 to 10

Validator Prompt:
Compute: 6 * 10 + 4 * 6 = 84
Answer (True/False): 6 * 10 = 60; 4 * 6 = 24; 60 + 24 = 84 = RHS || True

Compute: 2 * 8 + 4 * 17 = 33
Answer (True/False): 2 * 8 = 16; 4 * 17 = 68; 16 + 68 = 84 != RHS || False
```

## H  VALIDATOR EVALUATION

We report the validator accuracy in Table 5, and the groundtruth validator data is obtained as follows:

**Arithmetic and PlanArith**  We compute the right answer by evaluating the mathematical expression. We obtain and incorrect answer by prompting text-davinci-003 to output an incorrect answer and verify that the answer is indeed incorrect.

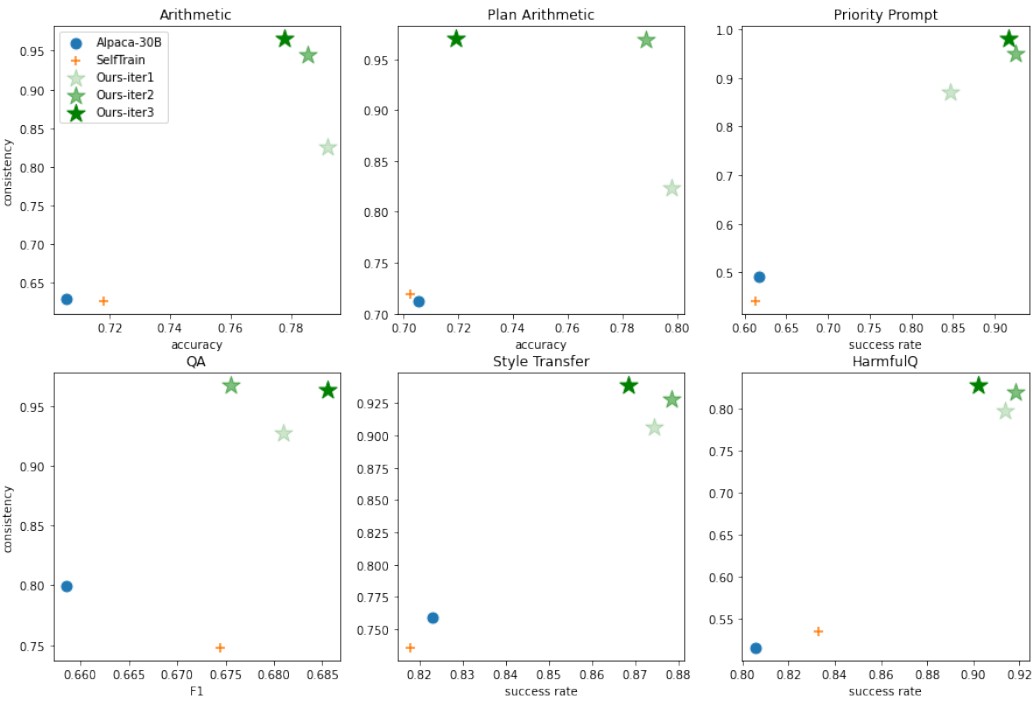

Figure 5: Correctness and Consistency scores are plotted jointly. We find that all iterations of consistency fine-tuning Pareto dominate the Alpaca and SelfTrain baselines.

**HarmfulQ and PromptP** We generate generator outputs by prompting text-davinci-003. We choose not to evaluate the validator on generations from Alpaca models because the text-davinci-003 generations tend to be higher quality and this also helps avoid evaluating on a distribution that's unfairly similar to the training dataset collected for the consistency fine-tuning pipeline. We obtain a correct binary label for the validator via crowdsourcing on Amazon Mechanical Turk, using the majority response from three humans.

**QA** We evaluate the validator on examples from the TriviaQA dataset. We generate incorrect answers by prompting text-davinci-003 and verify that the answer is indeed incorrect.

**Style** We evaluate the validator on the Yelp dataset from Li et al. (2018b). Each example includes a Yelp review stylized once positively, and once negatively.

## I JOINTLY CONSIDERING CORRECTNESS AND CONSISTENCY

In Table 3 and Table 5, we report consistency scores and factual correctness separately. Here, we consider the both metrics by plotting them simultaneously in Figure 5. For factual correctness, we use the mean of the generator/validator performance (e.g., accuracy for arithmetic and F1 score for QA). These plots more clearly show that all iterations of consistency fine-tuning Pareto dominate the base Alpaca and SelfTraining. Further, these plots indicate that the multiple iterations of consistency fine-tuning form a frontier in many tasks, allowing users of consistency fine-tuning to choose between trading off between additional improvements in consistency vs. factual correctness. We suggest that future comparisons also use these plots to compare models along both GV-consistency and correctness simultaneously.

## J EXTRAPOLATION

To examine the extrapolation performance of our consistency finetuned model, we construct the extrapolation evaluation data for three tasks: harmful questions, QA, and style transfer.

**Style transfer.** For style transfer, we consider a new style as a new task. For example, at training time, the model is trained on sentiment transfer and formality transfer tasks; and at test time, we evaluate the LM on unseen tasks like transfering to unseen styles.

In our experiment, we use the following 40 styles for training: analytical, descriptive, formal, sophisticated, educational, reflective, imaginative, simplified, persuasive, satirical, eloquent, opinionated, vivid, inspiring, colloquial, whimsical, detailed, factual, academic, structured, journalistic, conversational, romantic, passionate, witty, punning, candid, philosophical, technical, thought-provoking, inspirational, authoritative, poetic, playful, optimistic, informative, exaggerated, informal, lyrical, logical. For the extrapolation experiment, we evaluate on 12 styles: motivational, lighthearted, humorous, evocative, wry, entertaining, experimental, engaging, creative, narrative, positive, and succinct.

**QA.** For training, we use the unlabeled questions from TriviaQA dataset (Joshi et al., 2017), and for the extrapolation experiment we evaluate on questions from Natural Questions (Kwiatkowski et al., 2019).

**Harmful questions.** We generate harmful questions by prompting `text-davinci-003` model for harmful questions on a given topics (e.g., environment, psychology, health, social, race, etc.) We split the full set of questions based on their topics and use half towards training and the remaining towards evaluation. Specifically, the training topics include race, society, stereotypes, legal, and toxicity, and the extrapolation topics include economy, environment, ethics, physical, and psychological.

## K  MORE RELATED WORKS

**Self-Critique of Language Models** Our generator-validator setup resembles the idea of a Generative Adversarial Network (GAN), where the generative model produces text, and the discriminative model checks whether the text sample comes from the empirical data distribution or from the generative model (Goodfellow et al., 2014). One key difference is that the GAN objective aims to optimize the generative model to produce text that's undetectable by the discriminative model, resulting in disagreement/inconsistency between the two models, whereas our GV-consistency aims to let the generator and validator be consistent with each other. Another related idea is ELECTRA (Clark et al., 2020), a pre-training procedure that consists of a collaborative generator and discriminator. The generator replaces some tokens with plausible alternatives, and the discriminator predicts whether a token has been replaced or not. The optimal generator-discriminator pair would reach an agreement with each other. Our approach also aims to find agreement between a generator and a validator, but we focus on improving downstream task consistency (e.g., math, QA), unlike the representation learning goal of ELECTRA.

The most similar to our work is Constitutional AI (Bai et al., 2023), which prompts the base LM to generate responses to harm-inducing prompts, and then prompts the LM with a set of principles (e.g., harmlessness) to critique the generated responses. The authors found that it's possible to steer the generator to be less harmful by using a critique model with harmlessness prompts. Our work differs in two ways: First, we inject the same principle in both the generator and the validator, thus our approach can be regarded as self-critique for consistency; Second, we show gains on a wide range of tasks beyond harmlessness, like instruction following and arithmetic reasoning.

