# OpenReview forum: "Benchmarking and Improving Generator-Validator Consistency of Language Models"
_ICLR.cc/2024/Conference — ICLR 2024 poster_

### Official Review · Reviewer_pfQS · 2023-10-31

**Soundness:** 2 fair
**Presentation:** 2 fair
**Contribution:** 2 fair
**Rating:** 6
**Confidence:** 4

**Summary:**

This paper focus on a notable inconsistency issue in large language models,  introducing a framework for measuring generator-validator consistency. Then the paper proposes consistency fine-tuning approach, which significantly boosts the GV-consistency of Alpaca-30B from 60% to 93%. Moreover, this fine-tuning enhances both the quality of generation and the accuracy of validation without relying on labeled data.

**Strengths:**

1. The work introduces a novel and practical framework for assessing the consistency between generation and validation in language models. This framework addresses a critical gap in LM evaluation, providing a systematic approach to measure how well a model's generated responses align with its validation outputs.
2. The consistency fine-tuning approach is efficient way to improves GV-consistency and performance.

**Weaknesses:**

Dataset Construction: 1. The example provided in the Appendix from the task "Prompt Prioritization" appears to be an unsuitable choice for evaluating GV-consistency. This task specifically requests the model to enumerate the "benefits" of the 2nd amendment, which inherently biases the output towards positive aspects. Consequently, when the Validator is tasked with judging bias from the generated text, it is more likely to yield a biased ("B") outcome.
2. An evident drawback in the methodology is the absence of human evaluation for assessing the quality of data generated by Large Language Models (LLMs).
3. The current format for the "Close-book QA" task seems to introduce unnecessary complexity. Instead of maintaining a straightforward approach, such as simply judging whether the provided answer is correct or not, the task involves considering misleading answers. This inclusion of potentially misleading responses adds layers of uncertainty and can compromise the clarity of the evaluation process.  Evaluation Framework:
1. The results presented in Table 5 are perplexing, particularly regarding how the validator is evaluated for tasks like "PriorityP" and other LLM-generated tasks. The source of the "answer" used to judge the validator's performance is unclear, leading to confusion in interpreting the results.
2. The current framework predominantly focuses on consistency, with limited attention to accuracy, highlighted mainly in Table 5. It would be more beneficial to integrate factual correctness into the assessment, alongside consistency. Developing a leaderboard that encapsulates both these aspects would provide a more comprehensive and meaningful evaluation.

Writing issue: 1. Under Figure1 there are too much space which disrupts the flow of the text. 2. The improvement for the performance of validator is “6.3&” in abstract, but “6.5%” in section 6.3

**Questions:**

see weakness

---

> ### Author Response · Authors · 2023-11-19
>
> We thank Reviewer pfQS for their review and feedback. Reviewer pfQS primarily raises concerns with the task and evaluation design. We have strengthened the paper by adding additional experiments based on pfQS’s suggestions and clarifying areas of misunderstanding. We summarize the changes to the paper below. Please let us know if there are any remaining questions or concerns. Thanks!
>
> **“Assessing the quality of [the generated] data.”**
>
> Based on pfQS’s and 5N9m’s comments, we have added new analysis of the generated data used in consistency fine-tuning to Appendix A of the paper. Specifically, we analyze the accuracy (or F1 score) of the generated data before and after consistency-filtering, i.e., what fraction of the generator / validator responses in the generated data are factually correct. We also analyze the consistency rates, i.e., what fraction of the data is GV-consistent. We use the delta in these performance metrics as indications of how much consistency-filtering impacts the quality of the data. We analyze the arithmetic and QA tasks, since these tasks have a known ground-truth answer, which allows for automatically computing generator / validator accuracy (or F1 score).
>
> |                          |   Accuracy on Consistency-Filtered Data  |   Accuracy on Pre-Filtered Data  |   Delta  |
> |--------------------------|------------------------------------------|----------------------------------|----------|
> |   Generator (Alpaca-7B)  |   42.7%                                  |   39.3%                          |   3.4%   |
> |   Validator (Alpaca-7B)  |   57.6%                                  |   40.1%                          |   17.5%  |
>
>
>
>
> |                           |   Accuracy on Consistency-Filtered Data  |   Accuracy on Pre-Filtered Data  |   Delta  |
> |---------------------------|------------------------------------------|----------------------------------|----------|
> |   Generator (Alpaca-30B)  |   83%                                    |   66%                            |   17%    |
> |   Validator (Alpaca-30B)  |   91%                                    |   72%                            |   19%    |
>
>
>
> In the arithmetic task, we find that Alpaca-30B generates middling-accuracy data before consistency filtering: the generated generator data has 66% accuracy, and the validator data has 72% accuracy. However, after filtering for the consistent data, the accuracy rises to 83% for the generator and 91% for the validator, indicating that consistency filtering can significantly improve quality. Additionally, (unsurprisingly) consistency filtering significantly increases the consistency rate of the generated data from 62.9% to 100%. Importantly, model size plays a role here, where the smaller Alpaca-7B model generates low-accuracy data before consistency filtering (only 39.3% for the generator and 40.1% for the validator), and consistency filtering brings a smaller, though still significant improvement in accuracy (increasing to 42.7% for the generator and 57.6% for the validator). Consistency filtering still significantly increases the consistency rate from 50.4% to 100%.
>
> In the QA task, we see a similar trend, where (1) consistency filtering improves performance metrics of the generated data and also the consistency rate; and (2) the final performance metrics are dependent on model size. Recall that QA tasks the generator with generating a correct and incorrect answer to a question and the validator must choose which answer is correct. In this task, we cannot compute accuracy of the validator on the generated data, since both answers from the generator may be incorrect. (Note that this differs from the results in Table 5, which uses examples from TriviaQA with ground-truth answers, whereas here we’re analyzing generated examples.) Hence, we analyze only the word-wise generator F1 score.
>
> For the Alpaca-30B model, we find that consistency filtering increases generator F1 score from 0.49 to 0.51; and the consistency rate from 79.9% to 100%. For the Alpaca-7B model, we find that again the base generator F1 score is lower, and consistency filtering increases it from 0.39 to 0.41. Consistency filtering again significantly increases consistency rate (from 68.5% to 100%).
>
> **Overall, these results indicate that consistency filtering can substantially improve data quality in both generator / validator performance metrics, as well as in consistency, though the filtered data is imperfect and still contains significant noise.** We further note that a significant advantage of the consistency-filtering approach is that it is *unsupervised and does not require human intervention* on the generated data.

---

> ### Author Response · Authors · 2023-11-19
>
> **Concerns with the PromptPrioritization task.**
>
> To contextualize this concern, we first detail PromptPrioritization. In PromptPrioritization, the generator is given a task of “write about the benefits of {TASK}” while adhering to a given persona. Half of the examples purposely include contradicting tasks and personas, where the correct behavior of the generator should be to refuse the task because it contradicts its persona, e.g., “I’m sorry, I am a {PERSONA}, and so I cannot write about the benefits of {TASK}.” These contradicting tasks and personas were obtained by prompting text-davinci-003 to generate statements that a persona would never write. Note that PromptPrioritization broadly tests GV-consistency: the generator is asked to follow a task under a persona, and the validator is asked if the generator achieved the task under the correct persona.
>
> Reviewer pfQS raises the concern that PromptPrioritization may not actually test GV-consistency, because the validator response that’s marked as GV-inconsistent may not actually imply inconsistencies between the generator and validator. This can arise, though rarely, in examples with contradicting tasks and personas, such as the one in the Appendix referenced by pfQS, where the persona is “an advocate for gun control” and the task is “write about the benefits of the 2nd amendment.” The persona and task are intended to be contradictory, but could simultaneously be satisfied with text like “the 2nd amendment is good because it helped establish precedence for amending the Constitution.” Given this text, the validator may predict that the persona is “an advocate for the 2nd amendment,” which is marked as GV-inconsistent, but does not necessarily imply inconsistency between the generator and validator. We agree that this can happen, but it occurs extremely rarely in the data — we manually inspected 40 examples with contradicting tasks and personas from PromptPrioritization and found that it was only possible to reasonably simultaneously satisfy both 22.5% of the time (11.25% of the total examples). The remaining 89.75% do appropriately test for GV-consistency, so **PromptPrioritization still tests GV-consistency**, albeit with some noise. Further, while it is technically possible to satisfy the 11.25% of near-contradicting examples, we find that language models do not do so in practice, and instead clearly contradict their given persona. We thank pfQS for raising this and have updated the paper to clarify this detail in Appendix E.
>
> **The format for close-book QA “introduces unnecessary complexity [over] simply judging whether the provided answer is correct or not.”**
>
> We agree that the suggested setup of “judging whether the provided answer is correct or not” is also reasonable, and we’ve added new experiments under this setup to Appendix E. The results are in the below table, and we find that under this setup, consistency fine-tuning still increases GV-consistency, while improving validator performance and barely changing generator performance. In the table, we use the notation X —> Y (delta) to denote a value X before consistency fine-tuning, Y after consistency fine-tuning, and delta as the improvement Y - X.
>
> However, we maintain that the original multiple choice setting in the paper of selecting the correct answer from a set of potentially misleading possibilities is quite natural, as well: It mirrors a common use case of language models, i.e., answering multiple choice questions. Furthermore, we include this multiple choice setting to add diversity to the experiments, as other tasks (e.g., Arithmetic) already consider validators that determine if the answer is correct or not.
>
> |                      | Generator                   |   Validator                 |   Consistency             |
> |----------------------|-----------------------------|-----------------------------|---------------------------|
> |   QA-MultipleChoice  |   0.663 —> 0.675 (+0.012)   |   0.654 —> 0.696 (+0.042)   |   79.9% —> 96.4%(+16.5%)  |
> |   QA-T/F (new)       |   0.663 —> 0.660 (-0.003)   |   0.661 —> 0.701 (+0.04)    |   51.3% —> 54.5% (+3.2%)  |

---

> ### Author Response · Authors · 2023-11-19
>
> **Considering “both [factual correctness and GV-consistency] would provide a more comprehensive and meaningful evaluation”**
>
> We thank pfQS for the suggestion. As pfQS notes, the submission reports both factual correctness (Table 5) and GV-consistency (Table 3), but does not simultaneously consider both. We have added new plots to Appendix I, which plot both factual correctness and GV-consistency simultaneously. For factual correctness, we use the mean of the generator and validator performance (e.g., accuracy for arithmetic and F1 score for QA). These plots more clearly show that all iterations of consistency fine-tuning Pareto dominate the base Alpaca model and self training. Further, these plots indicate that the multiple iterations of consistency fine-tuning form a frontier in many tasks, allowing users of consistency fine-tuning to choose between trading off between additional improvements in consistency vs. factual correctness. We suggest that future comparisons also use these plots to compare models along both GV-consistency and correctness simultaneously.
>
> **Confusion about the validator evaluation.**
>
> We apologize for the missing validator evaluation details, which we have added to Section 5 and Appendix H of the paper and detail below:
> - Arithmetic and PlanArith: We compute the right answer by evaluating the mathematical expression. We obtain an incorrect answer by prompting text-davinci-003 to output an incorrect answer and verify that the answer is indeed incorrect. We evaluate the validator by randomly sampling between the correct and incorrect answer and asking the validator if the provided answer is correct or not.
> - HarmfulQ and PromptP: We generate generator outputs by prompting text-davinci-003. We choose not to evaluate the validator on generations from Alpaca models because the text-davinci-003 generations tend to be higher quality and this also helps avoid evaluating on a distribution that’s unfairly similar to the training dataset collected for the consistency fine-tuning pipeline. We obtain a correct binary label for the validator via crowdsourcing on Amazon Mechanical Turk, using the majority response from three humans.
> - QA. We evaluate the validator on examples from the TriviaQA dataset. We generate incorrect answers by prompting text-davinci-003 and verify that the answer is indeed incorrect.
> - Style. We evaluate the validator on the Yelp dataset from Li et al., 2018. Each example includes a Yelp review stylized once positively, and once negatively.
>
> **Writing concerns:**
> We have also updated the paper to fix the formatting issues raised by the reviewer and inconsistency between the abstract and Section 6.3, which both had a stale number.

---

> > ### Comment · Reviewer_pfQS · 2023-11-22
> >
> > Thanks for the authors' new experiments and detailed explanation. I am quite satisfied with the results but a bit worried about so many updates in the revised paper. So, I will increase my score to 6.

---

### Official Review · Reviewer_5N9m · 2023-11-01

**Soundness:** 4 excellent
**Presentation:** 4 excellent
**Contribution:** 4 excellent
**Rating:** 8
**Confidence:** 5

**Summary:**

- This paper proposed GV-consistency, an evaluation framework to measure the consistency between a LLM's generation and validation capabilities. Across arithmetic, QA, instruction following, and style transfer tasks, they surprisingly find even state-of-the-art LMs like GPT-4 have only 76% GV-consistency.
- To improve consistency, they propose "consistency fine-tuning" method that first filters the generator-validator response pairs to only keep consistent examples, and finetunes the LM to maximize likelihood of the consistent pairs.
- Through iterative fine-tuning, they significantly boost GV-consistency of Alpaca-30B from 60% to 93% on average across 6 diverse tasks.
- In terms of task performance, the fine-tuning even improves generator quality by 14% and validator accuracy by 6.3% without any labeled data.

**Strengths:**

- Discover and investigate the important issue of consistency between language model generation and validation.
- Thorough experiments across diverse tasks highlight the prevalence of inconsistency issues even for large models.
- Simple yet effective consistency fine-tuning method that requires no labels and meaningfully improves not only the consistency but also the task performance of the generator/validator. It can also transfer gains to new domains.

**Weaknesses:**

- Sometimes consistency fine-tuning fails to bootstrap generator or validator performance to beat the SelfTrain baseline. For example in Figure 5 (Alpaca-7B) and Table 5 (QA) (Alpaca-30B). The authors made a reasonable guess that the consistency filtered can be still of lower quality. If the authors can provide further analysis of the filtered data, it would be really helpful for us to fully understand the reason and figuring out how to avoid this kind of situation when applying the method to new tasks/datasets.

**Questions:**

- Figure 5 is a table, why the caption says Figure 5?

---

> ### Author Response · Authors · 2023-11-19
>
> We thank reviewer 5N9m for their thoughtful review and helpful suggestion. 5N9m suggests analyzing the consistency finetuning data to understand when it may help the generator/validator. We add this analysis to Appendix A of the paper and summarize below. Please let us know if there are any follow-up questions. Thanks!
>
> **Analysis of the consistency-filtered data.**
>
> We analyze the generator/validator performance of the generated consistency fine-tuning data, e.g., what fraction of the generator/validator responses in the generated data are factually correct. We consider the performance before and after filtering for consistency (i.e., the fine-tuning data for the SelfTrain baseline and the fine-tuning data for consistency fine-tuning respectively). We analyze the arithmetic and QA tasks, since these tasks include a known ground-truth answer, allowing for automatically computing generator/validator performance.
>
> **Overall, our analysis suggests that consistency fine-tuning improves generator performance exactly when the consistency-filtered data has much higher generator performance than the pre-filtered data, and analogously for the validator**. When the gap between pre- and post-filtered data is smaller, consistency fine-tuning also appears to improve the generator/validator less.
>
> The table below contains the results for Arithmetic on the Alpaca-7B model. We find that generator accuracy is only 3.4% higher on the consistency-filtered data than on the pre-filtered data. However, we find that validator accuracy is significantly higher (17.5%) post- vs. pre-filtering. Additionally, consistency finetuning barely changes the generator accuracy, slightly decreasing from 39% to 37%, but significantly improves validator accuracy, improving from 54% to 62%, suggesting that larger improvements from pre- to post-filtering translate to larger improvements after fine-tuning.
>
> |                          |   Acc on Consistency-Filtered Data  |   Acc on Pre-Filtered Data  |   Delta  |
> |--------------------------|------------------------------------------|----------------------------------|----------|
> |   Generator (Alpaca-7B)  |   42.7%                                  |   39.3%                          |   3.4%   |
> |   Validator (Alpaca-7B)  |   57.6%                                  |   40.1%                          |   17.5%  |
>
> We see a similar trend for the Alpaca-30B model for Arithmetic in the below tables. Both generator and validator accuracies are significantly higher post-filtering. We also find that consistency fine-tuning significantly improves both generator and validator accuracies over the base model and SelfTrain.
>
> |                           |   Acc on Consistency-Filtered Data  |   Acc on Pre-Filtered Data  |   Delta  |
> |---------------------------|------------------------------------------|----------------------------------|----------|
> |   Generator (Alpaca-30B)  |   83%                                    |   66%                            |   17%    |
> |   Validator (Alpaca-30B)  |   91%                                    |   72%                            |   19%    |
>
> |                       |   Consistency Fine-tuning  |   SelfTrain  |   Base Alpaca-30B  |
> |-----------------------|----------------------------|--------------|--------------------|
> |   Generator Acc  |   82.9%                    |   74.5%      |   74.3%            |
> |   Validator Acc  |   72.7%                    |   69.1%      |   66.8%            |
>
> We observe similar trends in QA. Recall that in QA, the generator is tasked with answering a question with a correct and incorrect answer, and the validator must choose which answer is correct. For the generated data, validator accuracy is ill-defined, since both answers from the generator may be incorrect. Hence, we only analyze the word-wise F1 score of the generator, and no performance metric for the validator. (Note that Table 5 in the submission computes validator accuracy on QA by using examples from TriviaQA with a ground truth correct answer and a generated incorrect answer, in contrast to the generated data, which may not have the correct answer).
>
> The generator F1 scores for both the 7B and 30B models barely change between pre- and post-consistency filtering. This may explain the results in Table 5, where consistency fine-tuning does not improve generator F1 compared to the base model or SelfTrain.
>
> |                    |   F1 score on Consistency-Filtered Data  |   F1 score on Pre-Filtered Data  |   Delta  |
> |--------------------|------------------------------------------|----------------------------------|----------|
> |   Generator (7B)   |   0.51                                   |   0.49                           |   0.02   |
> |   Generator (30B)  |   0.41                                   |   0.39                           |   0.02   |
>
> Importantly, even when consistency fine-tuning does not improve generator or validator performance, it still improves GV-consistency.

---

> > ### Comment · Reviewer_5N9m · 2023-11-20
> > **Thanks for the reply**
> >
> > Thanks for the additional experiments. The results seem reasonable to me. The performance improvement directly depends on the accuracy of the filtered data. It will be interesting to see in future work whether we can further filter the data when consistency filtering fails, such as in the case of generator Alpaca-7B, without the need for labeled data.
> >
> > Now I am satisfied with the author's response. I will keep my score unchanged.

---

### Official Review · Reviewer_bsD6 · 2023-11-04

**Soundness:** 2 fair
**Presentation:** 2 fair
**Contribution:** 2 fair
**Rating:** 6
**Confidence:** 5

**Summary:**

This paper basically proposes a simple data filtering approach for fine-tuning LLMs, by filtering out generated fine-tuning data that are inconsistent between generator and validator. The hypothesis is that when generator and validator agree, the generated data tend to be accurate. The paper defines Generator-Validator consistency (GV-consistency) score and finds that even powerful LLMs such as GPT-4 still suffer from a low to moderate GV-consistency. After fine-tuning LLMs with GV-consistency filtered data, the GV-consistency improves for the original ALPACA-30B model across 6 tasks and also outperforms baseline SELFTRAIN.  The paper also finds that GV-consistency fine-tuning improves consistency on unseen domains and tasks, and improves the generator and validator performance on the majority of tested tasks (except for QA and PlanArith).

**Strengths:**

(1)	The paper observes and defines GV-consistency issue and finds that even SOTA LLM  GPT-4 still suffers from a moderate GC-consistency score 76%.

(2)	The proposed approach is very simple. By generating and filtering GV-consistent data from LLMs and using the data for fine-tuning, the approach could improve GV-consistency of LLM and this procedure can be conducted iteratively. The paper observes gains on GV-consistency from 3 iterations across 6 tasks. This approach also could potentially improve generator and validator performance when the generator and validator have complementary strengths, hence could improve them without labeled data.

**Weaknesses:**

(1)	The paper is very empirical and lacks theoretical depth.

a.	When pointing out the GV-consistency issue, the paper did not provide insights or explanations or analyses on why even powerful LLMs suffer from generator-validator inconsistency.

b.	When evaluating GV-consistency of several competitive LLMs in Table 2, the paper did not shed light on why GPT-3.5 outperforms GPT-4 and davinci-003 on GV-consistency score, as GPT-4 outperforms GPT-3.5 in most  LLM evaluations and  davinci-003 is much larger than GPT-3.5 in model size.

(2)	Empirical evaluations also lack deeper and more complete analyses and explanations:

a.	Section 7.1, when evaluating the proposed approach on a smaller ALPACA-7B model, the paper finds that consistency fine-tuning improves the consistency scores for the two tasks, but did not always improve generator or validator performance. The paper hypothesizes that this could be due to the relatively low performance of initial generator/validator. However,  with the reported generator/validator performance of the initial model, it is not clear how “low” the initial generator/validator performance could impact the effectiveness of this approach. It would be useful to investigate on LLMs of other different model sizes and initial generator/validator performances on various tasks, to gain more insights on the effectiveness of the approach on different initial generator/validator performances, which could also correspond to different categories of tasks.

b.	 Section 7.2 is quite superficially analyzed. CC-FT could be considered as providing both consistent and inconsistent data for the model to fine-tune, whereas the consistency fine-tuning only uses consistent data to fine-tune. Table 6 shows that consistency fine-tuning performs much better than CC-FT. Again, no insights or analyses are provided.

**Questions:**

Please check the comments and concerns raised under Weaknesses.

---

> ### Author Response · Authors · 2023-11-19
>
> We thank reviewer bsD6 for their time and review. Reviewer bsD6 primarily asks for additional insights and analyses into generator-validator consistency. We have added several new experiments and additional discussion in the paper to address these, summarized below. Please let us know if there are any additional questions or concerns, during the discussion period. Thank you!
>
> **How do “different model sizes and initial generator/validator performances” impact consistency and generator/validator performance after consistency fine-tuning?**
>
> To answer this question, we have added new experiments with the Alpaca-7B and Alpaca-13B models on additional consistency tasks. In the below table, we report the following four metrics before and after consistency fine-tuning: (1) the generator performance (i.e., accuracy for Arithmetic, PlanArithmetic; F1 score for QA; and success rate for PriorityP); (2) the validator performance (i.e., accuracy); (3) the mean of the generator and validator performances (mean GV performance); and (4) the GV-consistency. We use the notation X —> Y (delta) to denote a value X before consistency fine-tuning, Y after consistency fine-tuning, and delta as the improvement Y - X.
>
> We find that **consistency fine-tuning significantly improves consistency across all tasks and all model sizes**. We also find that **consistency fine-tuning improves the mean of generator and validator performance across all tasks and model sizes**, though it does not necessarily always improve generator or validator performance individually.
> Furthermore, the magnitude of improvements to the mean GV performance tends to increase with model size.
>
> We have added these new experiments to Appendix C of the paper.
>
> |                          |   Generator                  |   Validator                 |   Mean GV accuracy          |   GV-Consistency            |
> |--------------------------|------------------------------|-----------------------------|-----------------------------|-----------------------------|
> |   Arithmetic (7B)        |   0.393 —> 0.369 (-0.024)    |   0.539 —> 0.623 (0.084)    |   0.466 —>  0.496   (0.03)  |    0.504 —> 0.864 (0.36)    |
> |   Arithmetic (30B)       |   0.668 —> 0.715 (0.047)     |   0.743 —> 0.869 (0.126)    |   0.706 —> 0.792 (0.086)    |   0.629 —> 0.826   (0.197)  |
> |   Plan Arithmetic (7B)   |   0.146 —> 0.196 (0.05)      |   0.763 —> 0.727 (-0.036)   |   0.455 —> 0.461 (0.006)    |   0.529 —> 0.779 (0.25)     |
> |   Plan Arithmetic (30B)  |   0.441 —> 0.631   (0.19)    |   0.97 —> 0.965 (-0.005)    |   0.706 —> 0.798 (0.092)    |   0.712 —> 0.824 (0.112)    |
> |   QA (7B)                |   0.626 —> 0.504   (-0.122)  |   0.598 —> 0.753   (0.155)  |   0.612 —> 0.628 (0.016)    |   0.685 —> 0.756 (0.071)    |
> |   QA (30B)               |   0.663 —> 0.671 (0.008)     |   0.654 —> 0.691   (0.037)  |   0.658 —>  0.681 (0.023)   |   0.799 —> 0.928 (0.129)    |
> |   PriorityP (7B)         |   0.46 —> 0.50 (0.04)        |   0.844 —> 0.908  (0.064)   |   0.650 —> 0.704 (0.054)    |   0.583 —> 0.648 (0.065)    |
> |   PriorityP (13B)        |   0.69 —> 0.71 (0.02)        |   0.543 — > 0.632 (-0.089)  |   0.616 —> 0.671 (0.055)    |   0.457 —> 0.701   (0.244)  |
> |   PriorityP (30B)        |   0.892 —> 0.922 (0.03)      |   0.754 —> 0.827 (0.073)    |   0.823 —> 0.875 (0.052)    |   0.49 —> 0.87 (0.38)       |
>
>
>
>
>
> **“Why [do] even powerful LLMs suffer from [generator-validator] consistency?”**
>
> As noted in prior work (Elazar et al, 2021), the training objectives for large language models do not explicitly enforce forms of consistency, including GV-consistency. More specifically: maximizing likelihood of the pre-training data does not directly optimize for consistency; in fact, the pre-training data may even contain inconsistent information from subjective content, such as one restaurant review stating “Starbucks had tasty food” and another stating “does Starbucks have good food? No!” In principle, the instruction-following step of model training (e.g., instruction-tuning or RLHF) could help improve consistency via careful design of the reward model or instruction-tuning data. However, gathering GV-consistent data has not been a priority in current instruction-following training pipelines, as this work is the first to highlight this type of inconsistency. We have added this discussion to Section 3 of the paper.

---

> ### Author Response · Authors · 2023-11-19
>
> **“Why [does] GPT-3.5 outperform GPT-4 and davinci-003 on GV-consistency score?”**
>
> To shed light on this question, we analyzed the consistency scores of these models across our tasks. As shown in Table 2 of the submission (and reproduced in the table below), **GPT-3.5 achieves the highest average GV-consistency primarily due to the prompt safety task (i.e., PromptP)**, where it achieves the highest GV-consistency by a wide margin (79.6 vs. 52.0 and 68.0 for GPT-4 and davinci-003 respectively). If we exclude this task, **both GPT-4 and davinci-003 achieve higher average GV-consistency than GPT-3.5**.
> To understand GPT-3.5’s superior consistency in PriorityP, we qualitatively analyzed responses of all 3 models in this task. We find that GPT-3.5 frequently generates abstentions of the form: “As an agent with the following persona […], I cannot provide an answer to the question advocating for the benefit of […],” which leads to high GV-consistency, since it is easy for the validator to predict the correct persona with these responses. In contrast, neither GPT-4 nor davinci-003 generate any abstentions, leading to lower GV-consistency in these tasks.
>
> The RLHF data for these models is not public information, so it is difficult to identify the cause of GPT-3.5’s abstention behavior in PriorityP with certainty. However, we speculate that it may arise from GPT-3.5 receiving more safety or prompt injection fine-tuning data to more conservatively accommodate its earlier public release via ChatGPT, though we do not know why GPT-4 does not exhibit similar abstention behavior. We have added this analysis and discussion to Appendix B of the paper.
>
> |                |   Arithmetic  |   PlanArith  |   PriorityP  |   QA    |   Style  |   Average  |   Average excluding PriorityP  |
> |----------------|---------------|--------------|--------------|---------|----------|------------|--------------------------------|
> |   GPT-3.5      |   67.7        |   66.0       |   79.6       |   89.6  |   92.6   |   79.1     |   79.0                         |
> |   GPT-4        |   75.6        |   62.0       |   52.0       |   95.3  |   94.3   |   75.8     |   81.8                         |
> |   davinci-003  |   84.4        |   60.0       |   68.0       |   86.9  |   85.7   |   77.0     |   79.3                         |
>
>
>
> **Why does “consistency fine-tuning perform much better than CC-FT?”**
>
> The trend that data filtering (used by consistency fine-tuning) performs better than CC-FT is not new in this paper. For example, Dubois et al. 2023, observe that data filtering via BinaryFeedME outperforms binary reward conditioning (i.e., CC-FT). The exact reason behind this performance gap is still unknown, though we speculate that it may stem from CC-FT using up some of its capacity to model multiple classes, while models trained on filtered data do not need to model the classes that have been filtered out. The key insight behind consistency fine-tuning is that fine-tuning data for GV-consistency can be generated *without human supervision* — and this insight is agnostic to the exact choice of fine-tuning method on this data. Hence, we believe that detailed analysis comparing various fine-tuning methods to comprehensively answer this question lies out of scope of this paper. We have updated Appendix D to clarify this point.
>
> **Reference**:
>
> Yanai Elazar, Nora Kassner, Shauli Ravfogel, Abhilasha Ravichander, Eduard Hovy, Hinrich Schütze, and Yoav Goldberg. Measuring and Improving Consistency in Pretrained Language Models, 2021
>
> Yann Dubois, Xuechen Li, Rohan Taori, Tianyi Zhang, Ishaan Gulrajani, Jimmy Ba, Carlos Guestrin, Percy Liang, and Tatsunori B. Hashimoto. AlpacaFarm: A simulation framework for methods that learn from human feedback, 2023

---

> > ### Comment · Reviewer_bsD6 · 2023-11-22
> > **Thanks for the responses**
> >
> > Thanks the authors for addressing my questions and concerns. I raised my score to 6.

---

### Author Response · Authors · 2023-11-19
**Author Response Summary**

We thank all of the reviewers for their helpful comments and suggestions. We’ve responded to each review separately and believe that we have addressed all reviewer concerns and questions. Please let us now if there are any other questions or concerns during the discussion period. Thank you!

We have strengthened the submission with additional experiments, discussions, and clarification based on the reviewer comments (changes in blue). We summarize the main changes below:
- We have included new experiments investigating the effect of model size and initial generator / validator performance on final performance after consistency fine-tuning in Appendix C.
- We have included new analyses for (1) the generated data used in consistency fine-tuning pipeline in Appendix A; and (2) jointly considering consistency and downstream performance (e.g., accuracy) in Appendix I.
- We have included discussion about why current language models suffer from generator-validator consistency in Section 3.
- We have revised Section 5 and Appendix H to clarify the dataset and evaluation details.

---

### Meta-Review · Area_Chair_xxNg · 2023-12-06

**Metareview:**

This paper finds the inconsistency between generating and validating an answer is prevalent in language models (LMs), i.e., a violation of GV-consistency, and proposes a framework for measuring the GV-consistency of LLMs. To improve consistency, it proposes consistency fine-tuning. The effectiveness of consistency fine-tuning is validated across 6 tasks.

The problem investigated in this paper is interesting and the proposed framework is useful for future researches. The proposed consistency fine-tuning method is reasonable and effective.

Weaknesses: The work is very empirical and lacks theoretical depth. The dataset and experiments need more analyses and explanations.

**Justification For Why Not Higher Score:**

see the meta-review.

**Justification For Why Not Lower Score:**

see the meta-review.

---

### Decision · Program_Chairs · 2024-01-16

Accept (poster)